# Are Sustainable Development Policies Really Feasible? Focused on the Petrochemical Industry in Korea

**Yongrok Choi** [1], **Hyoung Seok Lee** [2,*] **and Ahmed Mastur** [3]

1   Department of International Trade, Inha University, Inharo100, Nam-gu, Incheon 402-751, Korea
2   East Asia Environment Research Center, Inha University, Inharo100, Nam-Gu, Incheon 402-751, Korea
3   Global e-governance Program, Inha University, Inharo 100, Nam-gu, Incheon 402-751, Korea
*   Correspondence: zard2303@naver.com

**Abstract:** Korea inaugurated an emission trading scheme (ETS) in 2015 for its ambitious target to reduce 37% greenhouse gas per 2030 business-as-usual level. This study examines the sustainable governance of the Korean petrochemical industry, one of the world's major emitters of greenhouse gas, with 55 firms participating in ETS. On the basis of the non-radial, non-parametric directional distance function, this study derives three types of efficiencies: greenhouse gas technical efficiency (GTE), pure technical efficiency, and scale efficiency. Using these indices, this study also provides information for benchmarking for the fast followers. The findings of this study reveal the following. First, petrochemical industry exhibits 63.5% ETS performance, on average, showing huge potential improvement. Second, by decomposing GTE value, this study provides information from the perspective of scale to find out the oversupply issues in some petrochemical firms. Lastly, benchmark information for each firm is provided to enhance its efficiency.

**Keywords:** Paris agreement; Korean ETS; greenhouse gas; petrochemical industry; non-radial DDF-DEA

## 1. Introduction

In 1995, the historic conference of the United Nations Framework Convention on Climate Change (hereafter, UNFCCC) inaugurated its first Conference of Parties (COP) meeting to find the best sustainable development methods worldwide. In 2015 in Paris, an agreement was reached at COP's 21st session to ensure that the global average increase in temperature above the pre-industrial levels will not exceed 2 °C. Moreover, the agreement also stipulated to continue efforts to limit the temperature increase to 1.5 °C above the pre-industrial level. These agreements recognized that the accumulated efforts to control the temperature control will significantly reduce the risks and effects of climate change. To achieve the goal of UNFCCC, all 195 member countries must make clear and measurable efforts for sustainable development worldwide [1].

South Korea (hereafter Korea) is one of the largest emitter countries, and thus, overcoming this international challenge is inevitable. Therefore, in 2009, the Korean government had set its own goals to reduce greenhouse gas (hereafter GHG) emissions by 30% from the business as usual (BAU) level by 2020. However, this plan was revised since the 21st session of the COP meeting held in Paris in 2015 (Paris agreement). As shown in Figure 1., in this treaty, Korea suggested a more long-term, yet a little loosened, responsibility to reduce 37% of GHG emissions from the BAU level until 2030.

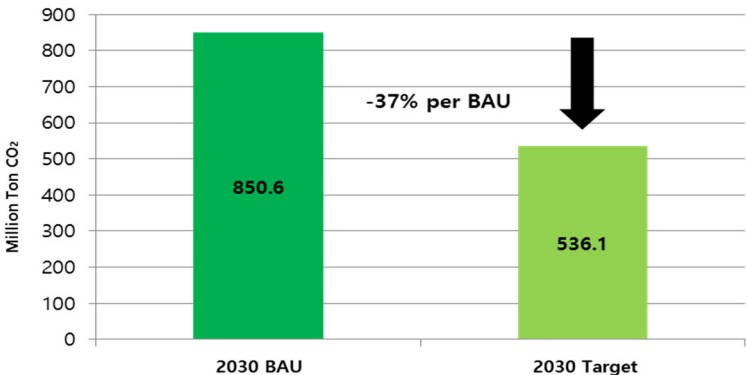

**Figure 1.** Korean business as usual (BAU) target by 2030 (Drawn by author).

To achieve this ambitious goal, Korean government has released an Emissions Trade Master scheme (2015–2025) in 2014. (The first phase of ETS started in 2015 and lasted until 2017. The second phase occupies a three-year-span from 2018 to 2020, and the final third phase is from 2021 to 2025. (Wikipedia, https://en.wikipedia.org/wiki/Emissions_Trading_Scheme_in_South_Korea)) However, despite this ambitious governmental effort, the carbon emission trading scheme (ETS) has been controversial. Most of the member firms under ETS argue that the government did not allocate sufficient permits and that the accuracy of interpretation of the uneven distribution is lacking. More than 40 cases have been filed against the government regarding the issue of permit allocation. Since ETS is an ongoing regime even in the future, firms need to find a synchronized way to maintain competency and environmental efficiency under the ETS regulation regime. Hence, utilizing data envelopment analysis (DEA), we aim to explore whether or not ETS influences sustainable performance of the Korean petrochemical industry.

We select the petrochemical industry in this study for the following reasons. First, petrochemical industry is not only one of the largest emitter industries in Korea but also has a considerable share of total GDP. As shown in Figure 2, the domestic production and export volume of petrochemical industry is much larger than its import, which implies its significant contribution to the national GDP. Second, out of the 525 firms, 85 firms participating in ETS are from the petrochemical industry, which has the largest share in ETS. Therefore, exploring the industry, with selective concentration of our research, will be the most insightful from the perspective of ETS feasibility.

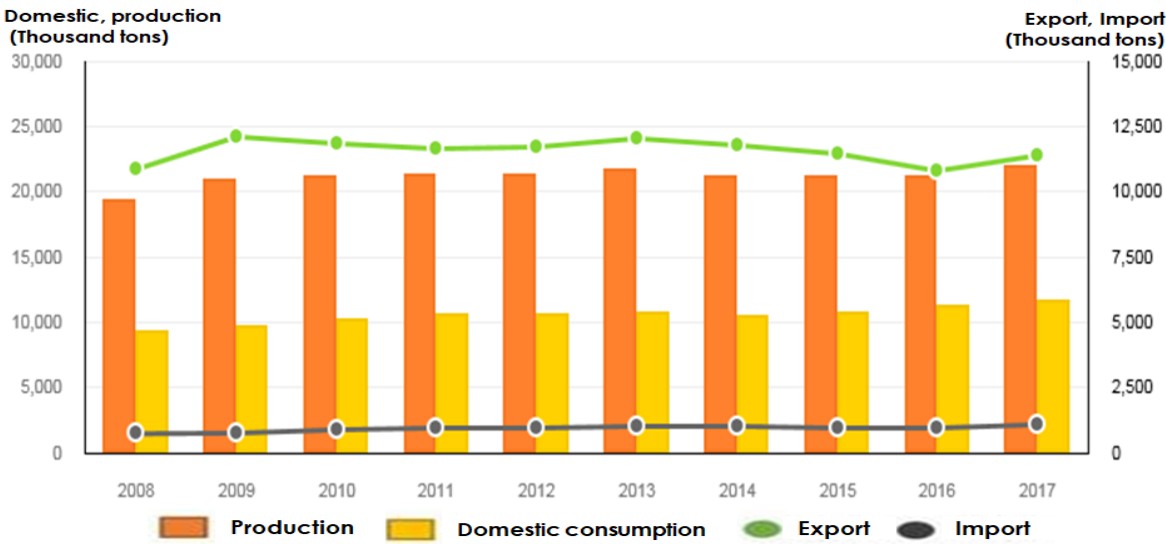

**Figure 2.** Statistic of petrochemical products. Source: KPIA, KITA (http://www.index.go.kr/potal/main/EachDtlPageDetail.do?idx_cd=1153).

Therefore, under this environmental regulation regime, firms should manage their operation from the perspective of environmental efficiency and find an optimal path to increase this efficiency, which is the main motivation of this study.

This study's unique contribution can be found in its data, methodology, and implications. First, in order to overcome the limitation of traditional DEA, we used directional distance function (DDF) with the more field-oriented firm-level data. Each datum is based on the precise and reliable source of a specific firm; hence, the quality of the study is enhanced in its practical implications. Second, this study explores not only each firm's performance but also trend by time as it covers seven consecutive years. Lastly, the petrochemical industry shows the largest share of ETS with 85 firms, representing 16% of the entire ETS members. Therefore, evaluating ETS performance in this sector would yield more reliable result.

The remainder of this paper is organized as follows. Section 2 discusses the literature background of sustainable development policies. Section 3 presents the methodology, and Section 4 shows the characteristics of data and empirical results. Section 5 concludes the paper with some political implications.

## 2. Literature Background of Sustainable Development Policies

Although numerous methodologies and models are available for environment-friendly sustainability, the distance function approach has obtained much popularity because it provides a more flexible analysis of the joint-production technology with good product and environmental bad outputs simultaneously [2]. Another advantage is that unlike the predetermined cost function, distance function does not require cost-specific data, which are comparatively difficult to obtain. Given only the quantitative data of multi-inputs and outputs, which are easier to acquire, several critical environmental production characteristics can be formally discussed, such as environmental technical efficiency, environmental productivity growth, the shadow prices of pollutants, and inter-factor substitution possibilities [3].

For environmental research, two types of distance functions are widely used in the literature. They are Shephard distance function [4] and DDF [5] respectively. In the Shephard distance function, although the multi-input/output analysis provides a basic logical frame for treatment, it is limited because of its proportional characteristics, which yield inappropriate output ratios, implying reduction of unnecessary output without any loss of optimized output [4]. As a result, most directional function experiments use DDF with more appropriate directional weights toward more good outputs and less bad outputs [5]. By using distance function models, more than 100 studies have resolved energy efficiency and the shadow prices of manufactured emissions [3].

In general, most early studies on the DDF were based on a radial model that estimates efficiency proportionately for all outputs [5]. However, this radial approach has limitation from two aspects. First, when the slack exists, the efficiency of the radial system can be highly biased [6]. Second, the radial model is based on the comparatively weak discrimination power in assessing the rank of firms [7]. Third, a radial efficiency measurement cannot measure energy-efficiency, such as single-factor efficiency, because DDF can only provide a similar rate of inefficiency [8]. These limitations have caused most recent studies to adopt non-radial DDF, which considers the non-radial slack instead of the conventional DDF [6,9–11].

Indeed, a numerous studies in the environment and energy (E&E) field widely adopt non-radial DDF, especially, those that utilize large emitter sectors as sample, such as power plants [12–17], steel and iron sectors [18–21], and other sectors, namely, chemical, cement, and ceramics industries [22–27]. Non-radial DDF approach is necessary because these sectors emit a considerable amount of undesirable output (GHG, including $CO_2$), which could be an obstacle to evaluate accurate performance. Petrochemical industry is one of the biggest emitters in Korea; hence, it is appropriate to adopt the non-radial DDF.

## 3. Methodology

*Greenhouse Gas Technical Efficiency and Decomposition*

For the non-radial DDF, it is essential to define "environmental production technology." In this study, we used three inputs ($x$): capital ($k$), labor ($l$), and energy consumption ($e$). These inputs produced a desirable output sales turnover ($y$) and undesirable output GHG ($b$) emissions. The production technology can be expressed as follows:

$$T = \{(x,y,b) : x \text{ can produce } (y,b)\} \tag{1}$$

where $T$ is often considered to be a general functional form of the production technology, given that "inertia is always possible," and "limited quantity of input can only produce limited quantities of output." Moreover, inputs and desired outputs are often supposed to be freely disposable. For the controlled environmental technology, $T$ must be applied for disadvantages [28]. First, the weak disposable assumption indicates that mitigating GHG (undesirable output) on the production process is costly in terms of proportional reduction of the desirable products. Second, null-jointness is essential for the production of GHG emissions; hence, the only way to eliminate GHG is to stop the production process. Mathematically, these two assumptions can be derived as follows:

(i) If $(K,L,E,T,C) \in T$ and $0 \le \theta \le 1$, then $(K,L,E,\theta T,\theta C) \in T$,

(ii) If $(K,L,E,T,C) \in T$ and $C=0$, then $T = 0$.

Following Zhou et al. [9], we can obtain $T$ for the $N$ regions as follows:

$$T_1 = \{(x,y,b) : \quad \sum_{n=1}^{N} z_n x_{mn} \le x_m, m = 1, \ldots, M,$$
$$\sum_{n=1}^{N} z_n y_{sn} \ge y_s, s = 1, \ldots, S, \tag{2}$$
$$\sum_{n=1}^{N} z_n b_{jn} = b_j, j = 1, \ldots, J,$$
$$z_n \ge 0, n = 1, \cdots, N\},$$

where $Z_n$ is a strength variable that can be employed to develop environmental technology by using a convex combination. In this study, we used both the constant returns to scale (CRS) and various return to scale (VRS) for $T$ to acquire diverse scale flexibility.

According to Zhou et al. [9], the formal non-radial DDF defined with unwanted output as follows:

$$\overrightarrow{D}(x,y,b;g) = sup\{\boldsymbol{w^T\beta} : ((x,y,b) + g \cdot diag(\boldsymbol{\beta})) \in T\}, \tag{3}$$

where $\boldsymbol{w} = (w_m^x, w_s^y, w_j^b)^T$. The number of inputs and outputs explain a normalized weight vector, $g = (-g_x, g_y, -g_b)$ is an evident directional vector, and $\beta = (\beta_m^x, \beta_s^y, \beta_j^b)^T \ge 0$ explain the vector of scaling factors. The measure of $\overrightarrow{D}(x,y,b;g)$ can be calculated with model (4):

$$
\begin{aligned}
\vec{D}^r(x, y, b; g) \quad &= \max w_m^x \beta_m^x + w_s^y \beta_s^y + w_j^b \beta_j^b \\
\text{s.t.} \quad &\sum_{n=1}^{N} z_n x_{mn} \leq x_m - \beta_m^x g_{xm}, m = 1, \ldots, M, \\
&\sum_{n=1}^{N} z_n y_{sn} \geq y_s + \beta_s^y g_{ys}, s = 1, \ldots S, \\
&\sum_{n=1}^{N} z_n b_{jn} = b_j - \beta_j^b g_{bj}, j = 1, \ldots J, \\
&z_n \geq 0, n = 1, 2, \cdots, N \\
&\beta_m^x, \beta_s^y, \beta_j^b \geq 0.
\end{aligned}
\tag{4}
$$

In this equation, the directional weight vector $g$ can be set up in different ways depending on the constraints. If $\vec{D}(x, y, b; g) = 0$, then the regions are located along the production technology frontier by $g$ direction. In this paper, vector is set as $g = (g_K, g_L, -g_E, g_T, -g_C)$, and weight vector (1/3, 1/3, 1/3, 1/3, 1/3).

Greenhouse gas technical efficiency (GTE) is acquired through non-radial DDF with the following variables. We set the weight vector of $S$ (desirable output), $J$ (undesirable output), and $M$ (three inputs of capital, labor, and energy). Moreover, we set directional vectors g $= (-x, y, -b)$ following that of Zhou et al. [9]. Suppose that $\beta_x^*, \beta_y^*$, and $\beta_b^*$ represent the best solution for Equation (4), then GTE can be obtained as

$$
GTE = 1 - \frac{1}{M + S + J}\left(\sum_{m=1}^{M} \beta_{xm}^* + \sum_{s=1}^{S} \beta_{ys}^* + \sum_{j=1}^{J} \beta_{bj}^*\right).
\tag{5}
$$

GTE is derived under CRS, whereas pure technical efficiency (PTE) is derived under the VRS condition. The assumption $Z_n = 1$ should be added to Equation (4) [29]. Once these two efficiencies are calculated, scale efficiency (SE) can be derived from the following equation:

$$
GTE = PTE \times SE.
\tag{6}
$$

Scale efficiency captures whether DMU uses scale effect well. The existence of unity indicates that the observation is in the highest productive scale size [30]. If SE is less than unity, it indicates that inefficiency exists, given the CRS technology. The scale inefficiency exhibited by DMU can be settled by either raising returns to scale (IRS) or reducing returns to scale (DRS) over time, to arrive at a higher efficiency. Based on this dynamic effect of production technology, the following DEA model can be introduced (7).

All DMUs can be evaluated at their initial stage of CRS using linear program. Suppose there exists $j(j = 1, 2, \ldots, n)$ DMUs and each DMU uses $i(i = 1, 2, \ldots, m)$ as its inputs to produce $r$ $(r = 1, 2, \ldots, s)$ as its outputs. Then, the efficiency from the production scale could be defined in its linear form:

$$
\min \theta \; S.T. : \; \theta x^i - X\lambda \geq 0, \; \theta y^i - Y\lambda \leq 0, \lambda \geq 0
\tag{7}
$$

In Equation (7), $x$ and $y$ are the vectors of DMU's inputs and outputs, respectively. Here, $\lambda$ is the weight vector. The function's value of $\theta$ is GTE. If $\theta = 1$, then the DMU is efficient on the frontier; if $\theta < 1$, then the DMU is inefficient, which uses more input $(1 - \theta)$, compared with the other DMUs. In addition, the return to scale (RTS) can be verified using the sum of calculated $\lambda$ values using the upper equation. If $\Sigma\lambda = 1$, then DMU is under the CRS condition, which indicates that both GTE and PTE are 1. Therefore, firms showing CRS status indicate that they have efficient input and output structure. By contrast, if the SE of firms is not equal to 1, then input and output structures need to be modified in order to achieve efficiency. In this case, when $\Sigma\lambda < 1$, DMUs are in DRS; whereas when $\Sigma\lambda > 1$, DMUs are in IRS.

## 4. Empirical Results

### 4.1. Data

In order to examine the environmental performance, we collected data from 55 Korean firms in the petrochemical industry from 2011 to 2017. Two output variables are selected: sales turnover (*T*) as the desirable output and GHG (*G*) as the undesirable output. For the input variables, we have selected two basic types of input, capital (*K*) and labor (*L*), and added energy (*E*) as the third input. We derived the data for labor, capital, and turnover from the Data Analysis, Retrieval, and Transfer System. Meanwhile, energy and GHG data have been collected from the GHG Inventory and Research Center in Korea. In general, in the fields of E&E research, pure $CO_2$ values were used with the macro type of data such as fuel cost rates, under the International Panel on Climate Change guidelines [13,31,32]. However, only the GHG information is available in Korea; hence, we focused on the firm-level data. Nonetheless, GHG data can be used as the proxy variable of $CO_2$ because carbon holds 80% of GHG emissions worldwide, which is similar in Korea according to the UNFCCC. Therefore, following Choi and Lee [1], we interpolate numerical values from GHG emissions data, which include methane (CH4), nitrogen (N2O), hydrofluorocarbons (HFCs), perfluorinated compounds (PFCs), and sulfur hexafluoride (SF6). The descriptive statistics for the data are shown in Table 1.

**Table 1.** Descriptive statistics.

| Variable | Type | Unit | Mean | Std. Dev. | Max. | Min. |
|---|---|---|---|---|---|---|
| Sales turnover | Desirable output | US$ billion | 1,566,278,032,416 | 2,630,226,310,547 | 1,721,714,5192,000 | 30,047,159,567 |
| GHG | Undesirable output | $CO_2$ equivalent tons | 539,525 | 1,090,403 | 5,979,058 | 14,079 |
| Capital | Input | US$ billion | 92,690,553,201 | 157,685,083,607 | 829,665,480,000 | 1,200,000,000 |
| Labor | Input | Per person | 935 | 1339 | 7619 | 29 |
| Energy | Input | Terajoules | 9830 | 20,808 | 115,303 | 256 |

Sources: Greenhouse Gas Inventory and Research Center of Korea (http://www.gir.go.kr/). DART: Data Analysis, Retrieval, and Transfer System (http://dart.fss.or.kr/).

As shown in Table 1, the standard deviation of each variable was huge since this study covered almost all petrochemical firms in Korea. Nonetheless, 55 firms belonged to the same industry, which implies there is no problem in heterogeneity. Therefore, the sample can be analyzed using DEA. As a last step, we determine the correlation of variables in order to verify the feasibility of the empirical result.

Table 2 presents the result of the correlation of five variables. As expected, two basic inputs (capital and labor) are significantly related with turnover because they are representative variables to explain production. However, these two inputs show no significance to GHG emission. By contrast, energy consumption shows high significance to both turnover and carbon emission. Especially, energy and GHG show a significant relationship. The result of this correlation matrix verifies the feasibility of the variable mix in this study and the appropriateness of analyzing the data in the perspective of energy and environment.

**Table 2.** Input and output correlation matrix.

| Variable | Capital | Labor | Energy | Turnover | Carbon |
|---|---|---|---|---|---|
| Capital | 1.000 | | | | |
| Labor | 0.250 | 1.000 | | | |
| Energy | 0.290 | 0.250 | 1.000 | | |
| Turnover | 0.372 | 0.548 | 0.765 | 1.000 | |
| Carbon | 0.297 | 0.256 | 0.990 | 0.747 | 1.000 |

*4.2. Empirical Result*

4.2.1. Greenhouse Gas Technical Efficiency (GTE)

The total GTE of petrochemical industry for seven consecutive years is calculated using Equations (4) and (5). The results are presented in Table 3, and Figure 3 shows annual GTE score's average trend. During the sample period, the GTE scores of 55 firms ranged from 0.5 to 1, and the average GTE score of them was 0.635, approximately 63.5%. This value supports two implications. First, although the average score was not quite high, the lowest value was greater than 50%, indicating that Korean petrochemical firms show good performance. Second, the remaining 36.5% efficiency enhancement in the petrochemical industry can still be obtained if the production is conducted on the environment-friendly frontier.

From the dynamic perspective over time, average GTE score spreads from 0.601 to 0.657. As shown in Figure 2, the highest score is obtained in 2011, whereas the lowest values are obtained in 2015 and 2016. However, an increasing trend was observed from 2017. This change occurred in the second stage of ETS; hence, the "Porter Hypothesis" is partially supported, which means that environmental regulation increase efficiency and encourage innovation for a more environment-friendly production process [33]. However, this positive change was limited only in 2017, and its sustainability in the future is not guaranteed. Hence, we should approach this question from the perspective of policy feasibility.

In general, there are three factors for the feasibility of any economic or business project: profitability, growth capacity, and stability. In these aspects, ETS may not be feasible right after implementation because of profitability. Although the core of this market-oriented regulation regime is beneficial by emission trading, unfortunately, it failed to draw firms' attention because of very low carbon price. According to the Korea Exchange, the average carbon price in January 2015 was less than US$10, which could not motivate firms to participate actively in ETS. In particular, even global firms such as Samsung were reluctant to join ETS and just paid the penalty. That is to say, ETS was perceived an added cost burden, not an opportunity for the firms. This negative situation seems to be reflected in the GTE for 2015 and 2016. Therefore, policy makers should find ways to motivate firms with profit-oriented policy. Once they succeed in motivating these firms, the number of participating firms will increase, and eventually, ETS stability can be achieved. This situation also reflects that firms are not proactively investing in green technology, resulting in the lack of growth capacity. Moreover, the Korean government randomly changed its environmental policy paradigm over time, resulting in the unpredictable, and thus, unstable conditions for the firms to decrease carbon emission actively. On the basis of these feasibility factors, we can conclude that the uptrend of efficiency may not be sustainable unless the Korean government proactively provides more transparent and optimal pathways toward the ambitious emission mitigation target by 2030.

**Table 3.** Gas technical efficiency (GTE) score calculated by directional distance function (DDF).

| Firms | 2011 | 2012 | 2013 | 2014. | 2015 | 2016 | 2017 |
|---|---|---|---|---|---|---|---|
| JW Life Science | 0.528 | 0.531 | 0.528 | 0.530 | 0.531 | 0.527 | 0.529 |
| KPX green chemical | 0.643 | 0.633 | 0.630 | 0.618 | 0.562 | 0.585 | 0.597 |
| LG MMA | 0.629 | 0.608 | 0.581 | 0.565 | 0.574 | 0.577 | 0.646 |
| OCI | 0.526 | 0.507 | 0.506 | 0.506 | 0.506 | 0.508 | 0.511 |
| SK Chemicals | 0.742 | 0.715 | 0.681 | 0.617 | 0.514 | 0.518 | 0.539 |
| SK Global Chemical | 1 | 0.953 | 0.961 | 0.989 | 0.808 | 0.744 | 0.807 |
| SK chemical | 0.529 | 0.518 | 0.522 | 0.513 | 0.509 | 0.512 | 0.500 |
| Ganggnam Hwasung | 0.621 | 0.611 | 0.629 | 0.627 | 0.609 | 0.614 | 0.665 |
| Kukdo chemical | 0.970 | 0.889 | 0.885 | 0.667 | 0.658 | 0.609 | 0.633 |
| Kumho Mitsui | 0.594 | 0.674 | 0.761 | 0.735 | 0.660 | 0.647 | 0.801 |
| Kumho Petrochemical | 0.630 | 0.611 | 0.589 | 0.575 | 0.546 | 0.533 | 0.554 |

**Table 3.** *Cont*.

| Firms | 2011 | 2012 | 2013 | 2014. | 2015 | 2016 | 2017 |
|---|---|---|---|---|---|---|---|
| Kumho tire | 0.551 | 0.553 | 0.546 | 0.546 | 0.540 | 0.538 | 0.537 |
| Kumho Polychem | 0.581 | 0.580 | 0.543 | 0.548 | 0.541 | 0.532 | 0.539 |
| Kumho P&B | 0.607 | 0.579 | 0.566 | 0.572 | 0.547 | 0.541 | 0.573 |
| Namhae Chemical | 1 | 0.961 | 1 | 0.920 | 0.808 | 0.733 | 0.811 |
| Nexen tire | 0.563 | 0.569 | 0.557 | 0.550 | 0.549 | 0.550 | 0.558 |
| Green cross | 0.707 | 0.701 | 0.68 | 0.693 | 0.701 | 0.665 | 0.691 |
| Daerim Ind. | 0.849 | 0.931 | 0.896 | 0.851 | 0.872 | 0.881 | 0.990 |
| Daesung Ind.gas | 0.504 | 0.504 | 0.504 | 0.505 | 0.515 | 0.505 | 0.505 |
| Daehan Chemicals | 0.634 | 0.638 | 0.627 | 0.630 | 0.584 | 0.570 | 0.583 |
| Lotte Chemical | 0.618 | 0.622 | 0.692 | 0.667 | 0.606 | 0.600 | 0.640 |
| Baekgwang Ind. | 0.541 | 0.540 | 0.526 | 0.503 | 0.508 | 0.508 | 0.509 |
| Samnam petrochemical | 1 | 0.913 | 0.865 | 0.789 | 0.703 | 0.706 | 0.729 |
| Samyang | 0.568 | 0.560 | 0.556 | 0.563 | 0.545 | 0.541 | 0.549 |
| SYC | 0.538 | 0.529 | 0.526 | 0.520 | 0.516 | 0.515 | 0.514 |
| Sundo chemical | 0.518 | 0.532 | 0.569 | 0.569 | 0.541 | 0.524 | 0.527 |
| Songwon Ind. | 0.692 | 0.696 | 0.688 | 0.698 | 0.689 | 0.703 | 0.699 |
| Yeocheon NCC | 0.701 | 0.710 | 0.702 | 0.685 | 0.597 | 0.584 | 0.615 |
| Yongsan chemical | 0.504 | 0.505 | 0.507 | 0.505 | 0.503 | 0.503 | 0.505 |
| Wooksung chemical | 0.538 | 0.539 | 0.541 | 0.536 | 0.535 | 0.537 | 0.544 |
| Unid | 0.535 | 0.520 | 0.521 | 0.521 | 0.520 | 0.511 | 0.519 |
| Youlchon chemical | 0.552 | 0.555 | 0.580 | 0.589 | 0.579 | 0.587 | 0.636 |
| ISU chemical | 0.737 | 0.765 | 0.751 | 0.698 | 0.592 | 0.576 | 0.600 |
| Jaewon Ind | 0.723 | 0.874 | 0.972 | 1 | 0.835 | 0.715 | 0.728 |
| CKD bio | 0.507 | 0.507 | 0.506 | 0.506 | 0.506 | 0.507 | 0.507 |
| KOC | 0.503 | 0.503 | 0.503 | 0.503 | 0.503 | 0.503 | 0.504 |
| Cosmo AM&T | 0.516 | 0.519 | 0.524 | 0.525 | 0.531 | 0.536 | 0.582 |
| Cosmo chemical | 0.506 | 0.507 | 0.508 | 0.507 | 0.506 | 0.506 | 0.508 |
| KOLON Ind. | 0.585 | 0.582 | 0.568 | 0.579 | 0.541 | 0.535 | 0.535 |
| Polymirae | 1 | 0.916 | 0.922 | 0.944 | 0.868 | 0.847 | 0.896 |
| Praxair | 0.503 | 0.504 | 0.505 | 0.507 | 0.509 | 0.510 | 0.511 |
| Filmax | 0.520 | 0.520 | 0.520 | 0.520 | 0.519 | 0.517 | 0.516 |
| Basf | 0.553 | 0.555 | 0.56 | 0.547 | 0.538 | 0.544 | 0.572 |
| Solvay | 0.597 | 0.587 | 0.590 | 0.579 | 0.571 | 0.575 | 0.591 |
| Korea stirolution | 1 | 0.957 | 0.981 | 0.915 | 0.816 | 0.787 | 0.898 |
| KEPITAL | 0.532 | 0.524 | 0.518 | 0.521 | 0.522 | 0.531 | 0.537 |
| Hansol Chemical | 0.525 | 0.523 | 0.524 | 0.516 | 0.516 | 0.518 | 0.518 |
| Hanil chemical | 0.750 | 0.697 | 0.714 | 0.734 | 0.746 | 0.734 | 0.848 |
| Hanhwa | 1 | 0.947 | 0.939 | 0.821 | 0.725 | 0.714 | 0.722 |
| Hanhwa Advanced materials | 0.805 | 0.800 | 0.631 | 0.626 | 0.623 | 0.627 | 0.627 |
| Hanhwa Chemical | 0.530 | 0.524 | 0.523 | 0.519 | 0.517 | 0.520 | 0.523 |
| HyundaiEP | 0.905 | 0.910 | 1 | 0.990 | 0.875 | 0.859 | 0.942 |
| Hwaseung R&A | 0.909 | 0.918 | 1 | 0.685 | 0.688 | 0.709 | 0.693 |
| Hwaseung Ind | 0.537 | 0.532 | 0.528 | 0.531 | 0.739 | 0.877 | 0.960 |
| Hyucamps | 0.648 | 1 | 0.807 | 0.713 | 0.607 | 0.594 | 0.633 |
| *Average* | 0.656 | 0.657 | 0.656 | 0.634 | 0.607 | 0.601 | 0.627 |

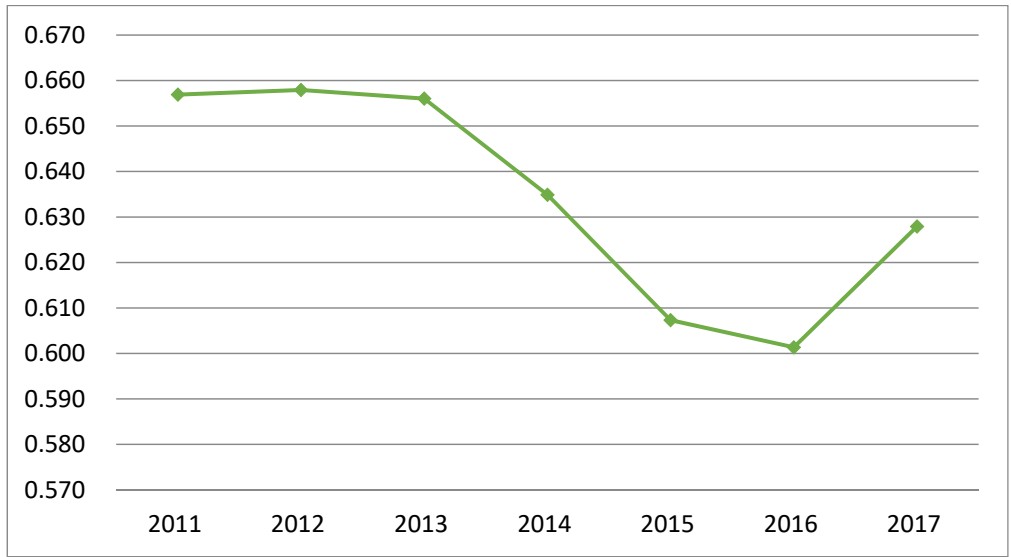

**Figure 3.** Gas technical efficiency (GTE) score from 2011 to 2017 (In average).

### 4.2.2. Decomposition of GTE and Result of Return to Scale

Following Equation (6), we decomposed GTE into PTE and SE. As mentioned above, PTE is the efficiency value obtained under the VRS condition, and SE is the ratio between two efficiencies score (PTE/GTE). The results obtained for both PTE and SE for the seven-year period are presented in Table 4. PTE scores are higher than the GTE scores, implying that the economies of scale coming from investment on the facilities may override the catching-up effect reflected in the low SE ratio.

Meanwhile, the average PTE score of Korean petrochemical industries is spread from 0.657 to 0.722 for seven consecutive years. As mentioned, PTE shows higher value than GTE because of SE; however, the average trend is almost similar to that of GTE, which indicates that SE does not affect the trend. Furthermore, the value differences between GTE and PTE vary. On the one hand, if two efficiency values are similar, then the two frontiers are adjacent. On the other hand, if there is huge gap between the two values, then the inefficiency that comes from scale-economy is severe. Explaining this deeper from the perspective of SE value, when SE is close to unity, we concurred that inefficiency was caused by the decrease in scale. If SE was close to zero, then the inefficiency caused by the scale was larger. Thus, the value of SE is important in determining firms' policies for scale effect. Thus, firms whose GTE and PTE were almost equal suggested that they used the "scale effect" well, which was an ideal condition. Otherwise, firms with low SE indicated that they established policy to bridge this gap from the perspective of scale-economy.

As a result, the average score of SE for the seven-year sample period is over 0.9, indicating that the petrochemical firms show high performance overall in using scale effect. However, some firms show relatively low SE, and they need to use the scale effect more effectively to increase their SE. Furthermore, it is essential for firms to bridge the gap between GTE and PTE to enhance GTE. As a solution, these firms should evaluate the feasibility of restructuring the intra-firms' input variable mix. We can provide suggestions for this restructuring by checking the DMU's RTS condition. Based on this RTS condition, each firm can make a specific plan on its input variable mix.

Returns to scale derived from Equation (7) are classified as CRS, DRS, and IRS. In this study, based on the 2017 data, 16 firms had the DRS condition, indicating that these firms should downsize their inputs to increase their GTE value. Meanwhile, 39 firms exhibited the IRS condition, implying that they needed to increase their inputs because in this condition, the output rate is higher than the inputs. This may stem from the fact that firms were influenced by the uptrend of GTE. That is to say, well-established ETS had a positive impact on firms and promoted their production. Meanwhile, the condition 'Constant' was not reported in 2017, indicating that no firms completely used the scale effect (GTE = PTE).

**Table 4.** Pure technical efficiency score (various return to scale (VRS)) and scale effect score.

| Firm's Name | 2011 | | 2012 | | 2013 | | 2014 | | 2015 | | 2016 | | 2017 | |
|---|---|---|---|---|---|---|---|---|---|---|---|---|---|---|
| | PTE | SE | PTE | SE | PTE | SE | PTE | SE | PTE | SE | PTE | SE | PTE | SE |
| JW Life Science | 0.949 | 0.557 | 0.962 | 0.551 | 0.903 | 0.584 | 0.807 | 0.656 | 0.770 | 0.689 | 0.713 | 0.739 | 0.711 | 0.744 |
| KPX green chemical | 0.867 | 0.742 | 0.744 | 0.851 | 0.704 | 0.895 | 0.678 | 0.912 | 0.608 | 0.924 | 0.631 | 0.927 | 0.645 | 0.926 |
| LG MMA | 0.639 | 0.984 | 0.618 | 0.983 | 0.590 | 0.984 | 0.573 | 0.984 | 0.583 | 0.984 | 0.586 | 0.984 | 0.653 | 0.988 |
| OCI | 0.540 | 0.974 | 0.509 | 0.996 | 0.507 | 0.996 | 0.508 | 0.996 | 0.507 | 0.997 | 0.510 | 0.996 | 0.521 | 0.981 |
| SK Chemicals | 0.813 | 0.913 | 0.765 | 0.934 | 0.718 | 0.948 | 0.709 | 0.871 | 0.568 | 0.906 | 0.570 | 0.908 | 0.589 | 0.914 |
| SK Global Chemical | 1 | 1 | 0.963 | 0.989 | 0.968 | 0.992 | 1 | 0.989 | 0.833 | 0.969 | 0.788 | 0.944 | 0.834 | 0.968 |
| SK chemical | 0.531 | 0.995 | 0.520 | 0.996 | 0.523 | 0.997 | 0.514 | 0.999 | 0.511 | 0.996 | 0.514 | 0.996 | 0.507 | 0.986 |
| Ganggnam Hwasung | 0.810 | 0.765 | 0.800 | 0.764 | 0.791 | 0.794 | 0.829 | 0.757 | 0.822 | 0.741 | 0.797 | 0.770 | 0.782 | 0.851 |
| Kukdo chemical | 1 | 0.970 | 0.911 | 0.975 | 0.904 | 0.979 | 0.670 | 0.996 | 0.658 | 0.999 | 0.610 | 0.998 | 0.634 | 0.999 |
| Kumho Mitsui | 0.619 | 0.958 | 0.697 | 0.966 | 0.786 | 0.969 | 0.761 | 0.966 | 0.690 | 0.957 | 0.676 | 0.957 | 0.822 | 0.974 |
| Kumho Petrochemical | 0.697 | 0.904 | 0.677 | 0.903 | 0.650 | 0.906 | 0.632 | 0.908 | 0.595 | 0.917 | 0.569 | 0.936 | 0.592 | 0.935 |
| Kumho tire | 0.559 | 0.987 | 0.563 | 0.982 | 0.546 | 0.999 | 0.546 | 0.999 | 0.541 | 0.997 | 0.539 | 0.997 | 0.538 | 0.996 |
| Kumho Polychem | 0.600 | 0.967 | 0.600 | 0.967 | 0.557 | 0.974 | 0.562 | 0.976 | 0.550 | 0.983 | 0.542 | 0.982 | 0.548 | 0.984 |
| Kumho P&B | 0.631 | 0.962 | 0.606 | 0.954 | 0.586 | 0.965 | 0.597 | 0.957 | 0.556 | 0.984 | 0.552 | 0.980 | 0.587 | 0.977 |
| Namhae Chemical | 1 | 1 | 0.962 | 0.999 | 1 | 1 | 0.921 | 0.998 | 0.809 | 0.999 | 0.733 | 0.999 | 0.818 | 0.991 |
| Nexen tire | 0.569 | 0.989 | 0.582 | 0.978 | 0.569 | 0.978 | 0.561 | 0.980 | 0.562 | 0.977 | 0.561 | 0.980 | 0.563 | 0.990 |
| Green cross | 0.872 | 0.810 | 0.844 | 0.830 | 0.772 | 0.879 | 0.777 | 0.891 | 0.773 | 0.907 | 0.688 | 0.966 | 0.715 | 0.967 |
| DaerimInd. | 0.911 | 0.932 | 1 | 0.931 | 0.977 | 0.916 | 0.915 | 0.930 | 0.934 | 0.933 | 0.952 | 0.925 | 1 | 0.990 |
| DaesungInd.gas | 0.507 | 0.993 | 0.506 | 0.994 | 0.506 | 0.996 | 0.506 | 0.996 | 0.517 | 0.997 | 0.506 | 0.997 | 0.505 | 0.998 |
| Daehan Chemicals | 0.634 | 0.999 | 0.638 | 0.999 | 0.627 | 0.999 | 0.630 | 0.999 | 0.585 | 0.998 | 0.573 | 0.995 | 0.584 | 0.999 |
| Lotte Chemical | 0.650 | 0.951 | 0.645 | 0.964 | 0.723 | 0.957 | 0.688 | 0.969 | 0.626 | 0.968 | 0.619 | 0.970 | 0.652 | 0.981 |
| Baekgwang Ind. | 0.549 | 0.985 | 0.546 | 0.989 | 0.530 | 0.991 | 0.517 | 0.973 | 0.526 | 0.964 | 0.524 | 0.968 | 0.525 | 0.969 |
| Samnam petrochemical | 1 | 1 | 0.914 | 0.998 | 0.870 | 0.994 | 0.795 | 0.992 | 0.710 | 0.990 | 0.716 | 0.986 | 0.738 | 0.987 |
| Samyang | 0.598 | 0.949 | 0.588 | 0.953 | 0.586 | 0.948 | 0.591 | 0.952 | 0.583 | 0.935 | 0.581 | 0.930 | 0.586 | 0.937 |
| SYC | 0.635 | 0.847 | 0.644 | 0.822 | 0.652 | 0.808 | 0.640 | 0.812 | 0.662 | 0.779 | 0.651 | 0.791 | 0.645 | 0.796 |
| Sundo chemical | 1 | 0.518 | 0.977 | 0.545 | 1 | 0.569 | 1 | 0.569 | 0.907 | 0.596 | 0.891 | 0.588 | 0.902 | 0.584 |
| Songwon Ind. | 0.693 | 0.999 | 0.697 | 0.998 | 0.690 | 0.996 | 0.701 | 0.996 | 0.691 | 0.996 | 0.706 | 0.995 | 0.703 | 0.994 |
| Yeocheon NCC | 0.718 | 0.976 | 0.723 | 0.981 | 0.717 | 0.979 | 0.702 | 0.975 | 0.626 | 0.954 | 0.610 | 0.957 | 0.640 | 0.961 |
| Yongsan chemical | 0.538 | 0.937 | 0.538 | 0.938 | 0.541 | 0.937 | 0.539 | 0.936 | 0.538 | 0.935 | 0.537 | 0.937 | 0.541 | 0.933 |
| Wooksung chemical | 0.874 | 0.615 | 0.935 | 0.576 | 0.846 | 0.639 | 0.958 | 0.559 | 0.932 | 0.573 | 0.938 | 0.573 | 1 | 0.544 |
| Unid | 0.536 | 0.998 | 0.521 | 0.998 | 0.522 | 0.999 | 0.522 | 0.999 | 0.521 | 0.998 | 0.511 | 0.999 | 0.520 | 0.998 |
| Youlchon chemical | 0.606 | 0.911 | 0.605 | 0.917 | 0.608 | 0.954 | 0.603 | 0.977 | 0.628 | 0.922 | 0.631 | 0.930 | 0.640 | 0.995 |
| ISU chemical | 0.769 | 0.959 | 0.810 | 0.944 | 0.796 | 0.942 | 0.735 | 0.949 | 0.611 | 0.968 | 0.582 | 0.988 | 0.621 | 0.966 |

**Table 4.** *Cont.*

| Firm's Name | 2011 | | 2012 | | 2013 | | 2014 | | 2015 | | 2016 | | 2017 | |
|---|---|---|---|---|---|---|---|---|---|---|---|---|---|---|
| | PTE | SE | PTE | SE | PTE | SE | PTE | SE | PTE | SE | PTE | SE | PTE | SE |
| Jaewon Ind | 0.794 | 0.910 | 0.897 | 0.974 | 1 | 0.972 | 1 | 1 | 0.848 | 0.983 | 0.726 | 0.985 | 0.741 | 0.982 |
| CKD bio | 0.550 | 0.922 | 0.545 | 0.929 | 0.548 | 0.924 | 0.547 | 0.925 | 0.546 | 0.928 | 0.549 | 0.924 | 0.544 | 0.931 |
| KOC | 0.583 | 0.861 | 0.592 | 0.849 | 0.600 | 0.839 | 0.602 | 0.836 | 0.610 | 0.825 | 0.614 | 0.820 | 0.597 | 0.843 |
| Cosmo AM&T | 0.589 | 0.877 | 0.638 | 0.814 | 0.643 | 0.815 | 0.661 | 0.795 | 0.748 | 0.710 | 0.689 | 0.778 | 0.638 | 0.913 |
| Cosmo chemical | 0.529 | 0.957 | 0.532 | 0.953 | 0.538 | 0.944 | 0.540 | 0.938 | 0.544 | 0.931 | 0.551 | 0.917 | 0.562 | 0.903 |
| KOLON Ind. | 0.618 | 0.946 | 0.611 | 0.952 | 0.595 | 0.955 | 0.615 | 0.941 | 0.555 | 0.976 | 0.548 | 0.976 | 0.548 | 0.977 |
| Polymirae | 1 | 1 | 0.928 | 0.987 | 0.935 | 0.986 | 0.953 | 0.990 | 0.875 | 0.991 | 0.867 | 0.976 | 0.917 | 0.977 |
| Praxair | 0.509 | 0.988 | 0.509 | 0.990 | 0.508 | 0.993 | 0.508 | 0.997 | 0.511 | 0.997 | 0.513 | 0.994 | 0.514 | 0.995 |
| Filmax | 0.659 | 0.789 | 0.663 | 0.784 | 0.670 | 0.775 | 0.676 | 0.768 | 0.682 | 0.761 | 0.676 | 0.764 | 0.669 | 0.772 |
| Basf | 0.625 | 0.884 | 0.634 | 0.875 | 0.643 | 0.870 | 0.618 | 0.885 | 0.579 | 0.929 | 0.601 | 0.904 | 0.664 | 0.862 |
| Solvay | 0.605 | 0.986 | 0.597 | 0.984 | 0.603 | 0.978 | 0.593 | 0.976 | 0.587 | 0.973 | 0.592 | 0.971 | 0.610 | 0.969 |
| Korea stirolution | 1 | 1 | 0.962 | 0.994 | 0.981 | 0.999 | 0.924 | 0.990 | 0.834 | 0.978 | 0.802 | 0.980 | 0.905 | 0.992 |
| KEPITAL | 0.544 | 0.979 | 0.535 | 0.979 | 0.528 | 0.982 | 0.530 | 0.982 | 0.530 | 0.983 | 0.540 | 0.983 | 0.544 | 0.986 |
| Hansol Chemical | 0.551 | 0.953 | 0.547 | 0.956 | 0.548 | 0.956 | 0.548 | 0.942 | 0.546 | 0.945 | 0.546 | 0.947 | 0.540 | 0.959 |
| Hanil chemical | 1 | 0.750 | 0.953 | 0.732 | 0.978 | 0.729 | 0.939 | 0.782 | 0.940 | 0.793 | 0.948 | 0.774 | 1 | 0.848 |
| Hanhwa | 1 | 1 | 0.954 | 0.992 | 0.956 | 0.982 | 0.965 | 0.851 | 0.975 | 0.744 | 0.881 | 0.810 | 0.769 | 0.938 |
| Hanhwa Advanced materials | 0.857 | 0.938 | 0.849 | 0.942 | 0.733 | 0.861 | 0.703 | 0.891 | 0.687 | 0.906 | 0.689 | 0.910 | 0.701 | 0.894 |
| HanhwaChemical | 0.596 | 0.888 | 0.569 | 0.920 | 0.563 | 0.928 | 0.556 | 0.934 | 0.549 | 0.941 | 0.561 | 0.928 | 0.574 | 0.910 |
| HyundaiEP | 0.992 | 0.912 | 0.938 | 0.970 | 1 | 1 | 0.991 | 0.999 | 0.876 | 0.998 | 0.861 | 0.997 | 0.947 | 0.994 |
| HwaseungR&A | 0.910 | 0.998 | 0.923 | 0.994 | 1 | 1 | 0.794 | 0.862 | 0.793 | 0.867 | 0.801 | 0.885 | 0.785 | 0.882 |
| Hwaseung Ind | 0.617 | 0.869 | 0.617 | 0.861 | 0.614 | 0.860 | 0.619 | 0.857 | 0.747 | 0.990 | 0.881 | 0.995 | 0.961 | 0.998 |
| Hyucamps | 0.678 | 0.956 | 1 | 1 | 0.894 | 0.901 | 0.795 | 0.896 | 0.627 | 0.968 | 0.611 | 0.971 | 0.643 | 0.984 |
| Average | 0.722 | 0.916 | 0.720 | 0.920 | 0.715 | 0.922 | 0.698 | 0.917 | 0.668 | 0.919 | 0.657 | 0.923 | 0.677 | 0.933 |

### 4.2.3. Benchmark Information

We have already determined solutions from the perspective of scale; however, setting a detailed input mix is still difficult for each firm. Fortunately, as mentioned, DEA provides the benchmark information for inefficient DMUs, which shows the GTE value less than unity. These firms can learn from a set of efficient DMUs, which are called the "reference set", that indicates that the efficiency value equal to unity of a firm makes them an efficient and ideal role models for other inefficient firms. That is to say, they are batched when they show similar input and output structure. Equation (8) explains how inefficient DMUs can approach unity. The λ value can be defined as the level of impact skilled DMUs give to each inefficient DMU. The result of the benchmarking information in this study is presented in Table 5. The main goal of benchmarking is providing a guide in organizing mix inputs in the future, not for analyzing trend; thus, we briefly provide information focusing on the sample year 2017.

$$Reference\ set's\ input * \lambda = inefficient\ DMU's\ input\ target \qquad (8)$$

**Table 5.** Firms for benchmark.

| Firms (2107) | RTS | Efficient DMU (Number of Reference Sets) | Benchmark (λ) |
|---|---|---|---|
| JW Life Science | Increasing | | Hanhwa 2011 (0.048561) |
| KPX green chemical | Increasing | | Namhae Chemical 2011 (0.093555); Samnam petrochemical2011 (0.032129); HyundaiEP2013 (0.257384) |
| LG MMA | Increasing | | Namhae Chemical2011 (0.283217); Samnam petrochemical 2011 (0.164708); HyundaiEP 2013 (0.301541) |
| OCI | Decreasing | | Namhae Chemical 2013 (1.317825); HyundaiEP 2013 (1.098502); HwaseungR&A 2013 (1.328494) |
| SK Chemicals | Increasing | Hanhwa2011 * 23 | Namhae Chemical 2011 (0.091120); Korea stirolution 2011 (0.116544) |
| SK Global Chemical | Decreasing | Namhae Chemical2011 * 17 Samnam Petrochemical2011 * 15 | SK Global Chemical 2011 (0.456638); Samnam petrochemical 2011 (2.005343) |
| SK chemical | Increasing | Korea stirolution2011 * 8 SK Global Chemical 2011 * 7 | Hanhwa 2011 (0.030053) |
| Ganggnam Hwasung | Increasing | Polymirae2011 * 5 Hyucamps2012 * 1 | Jaewon Ind 2014 (0.346036); HwaseungR&A 2013 (0.185762) |
| Kukdo chemical | Decreasing | Namhae Chemical2013 * 12 HyundaiEP2013 * 10 HwaseungR&A2013 * 24 | Namhae Chemical 2013 (0.277076); HyundaiEP 2013 (0.688459); HwaseungR&A 2013 (0.106776) |
| Kumho Mitsui | Increasing | Jaewon Ind2014 * 7 | Namhae Chemical 2011 (0.268066); Polymirae 2011 (0.327331); Korea stirolution 2011 (0.144913) |
| Kumho Petrochemical | Decreasing | | Namhae Chemical 2011 (2.282235); Samnam petrochemical 2011 (1.269622); Polymirae 2011 (0.460889) |
| Kumho tire | Increasing | | Hanhwa 2011 (0.747052) |
| Kumho Polychem | Increasing | | Namhae Chemical 2011 (0.359559); Samnam petrochemical 2011 (0.064860); HyundaiEP 2013 (0.102880) |
| Kumho P&B | Decreasing | | Samnam petrochemical 2011 (0.328405); Korea stirolution 2011 (1.157770) |

**Table 5.** *Cont.*

| Firms (2107) | RTS | Efficient DMU (Number of Reference Sets) | Benchmark (λ) |
|---|---|---|---|
| Namhae Chemical | Increasing | | Namhae Chemical 2011 (0.795987); Hanhwa 2011 (0.022958); HwaseungR&A 2013 (0.029142); Hyucamps 2012 (0.013084) |
| Nexen tire | Decreasing | | Jaewon Ind 2014 (0.359655); HwaseungR&A 2013 (1.643121) |
| Green cross | Increasing | | Hanhwa 2011 (0.100143); HwaseungR&A 2013 (0.640011) |
| Daerim Ind. | Decreasing | | SK Global Chemical 2011 (0.002074); Samnam petrochemical 2011 (0.005912); Jaewon Ind 2014 (4.507030); HwaseungR&A 2013 (6.350246) |
| Daesung Ind.gas | Increasing | | Namhae Chemical 2013(0.113909); Hanhwa 2011 (0.017673); HwaseungR&A 2013 (0.453252) |
| Daehan Chemicals | Decreasing | | SK Global Chemical 2011 (0.028732); Samnam petrochemical 2011 (0.652468); HwaseungR&A 2013 (0.572287) |
| Lotte Chemical | Decreasing | Hanhwa2011 * 23 Namhae Chemical2011 * 17 Samnam Petrochemical2011 * 15 Korea stirolution2011 * 8 SK Global Chemical 2011 * 7 Polymirae2011 * 5 Hyucamps2012 * 1 Namhae Chemical2013 * 12 HyundaiEP2013 * 10 HwaseungR&A2013 * 24 Jaewon Ind2014 * 7 | SK Global Chemical 2011 (0.582863); Samnam petrochemical 2011 (0.984766); HwaseungR&A 2013 (2.083099) |
| Baekgwang Ind. | Increasing | | Namhae Chemical 2013 (0.125532); Hanhwa 2011 (0.020293) |
| Samnam petrochemical | Increasing | | Namhae Chemical 2011 (0.061857); Samnam petrochemical 2011 (0.162388); Polymirae 2011 (0.554011) |
| Samyang | Increasing | | Namhae Chemical 2011 (0.287392); Korea stirolution 2011 (0.047018) |
| SYC | Increasing | | Hanhwa 2011 (0.031505) |
| Sundo chemical | Increasing | | Samnam petrochemical 2011 (0.003805); HyundaiEP 2013 (0.007622); HwaseungR&A 2013 (0.029718) |
| Songwon Ind. | Decreasing | | SK Global Chemical 2011 (0.008470); Jaewon Ind 2014 (2.375792); HwaseungR&A 2013 (0.124196) |
| Yeocheon NCC | Decreasing | | SK Global Chemical 2011 (0.075411); Samnam petrochemical 2011 (2.945498); HwaseungR&A 2013 (0.166235) |
| Yongsan chemical | Increasing | | Hanhwa 2011 (0.009686); HwaseungR&A 2013 (0.072694) |
| Wooksungchemical | Increasing | | Hanhwa 2011 (0.003512); HwaseungR&A 2013 (0.095427) |
| Unid | Increasing | | Namhae Chemical 2013 (0.383458); Hanhwa 2011 (0.061025); HwaseungR&A 2013 (0.073768) |
| Youlchon chemical | Increasing | | Jaewon Ind 2014 (0.453109); HwaseungR&A 2013 (0.417042) |
| ISUchemical | Decreasing | | Namhae Chemical 2011 (0.748763); Korea stirolution 2011 (0.530164) |

**Table 5.** *Cont.*

| Firms (2107) | RTS | Efficient DMU (Number of Reference Sets) | Benchmark (λ) |
|---|---|---|---|
| Jaewon Ind | Increasing | | Jaewon Ind 2014 (0.899241); HwaseungR&A 2013 (0.009052) |
| CKD bio | Increasing | | Hanhwa 2011 (0.031379); HwaseungR&A 2013 (0.038350) |
| KOC | Increasing | | Hanhwa 2011 (0.014351) |
| Cosmo AM&T | Increasing | | Namhae Chemical 2013 (0.135035); Hanhwa 2011 (0.064811) |
| Cosmo chemical | Increasing | | Hanhwa 2011 (0.040064) |
| KOLON Ind. | Decreasing | | Namhae Chemical 2013 (0.326343); Hanhwa 2011 (0.060752); HwaseungR&A 2013 (3.189857) |
| Polymirae | Increasing | | Namhae Chemical 2011 (0.121997); Samnam petrochemical 2011 (0.108101); Polymirae 2011 (0.495369) |
| Praxair | Increasing | | Namhae Chemical 2013 (0.570592); Hanhwa 2011 (0.011743) |
| Filmax | Increasing | | Hanhwa 2011 (0.034096) |
| Basf | Decreasing | Hanhwa2011 * 23 Namhae Chemical2011 * 17 Samnam Petrochemical2011 * 15 Korea stirolution2011 * 8 SK Global Chemical 2011 * 7 Polymirae2011 * 5 Hyucamps2012 * 1 Namhae Chemical2013 * 12 HyundaiEP2013 * 10 HwaseungR&A2013 * 24 Jaewon Ind2014 * 7 | Namhae Chemical 2011 (2.146136); Korea stirolution 2011 (0.615560) |
| Solvay | Increasing | | Namhae Chemical 2011 (0.239045); Namhae Chemical 2013 (0.183106); HyundaiEP 2013 (0.260430) |
| Korea stirolution | Increasing | | Namhae Chemical 2011 (0.128825); Korea stirolution 2011 (0.799330) |
| KEPITAL | Increasing | | Namhae Chemical 2011 (0.004619); Samnam petrochemical 2011 (0.045602); HyundaiEP 2013 (0.642860) |
| Hansol Chemical | Increasing | | Namhae Chemical 2013 (0.016250); Hanhwa 2011 (0.117349) |
| Hanil chemical | Increasing | | SK Global Chemical 2011 (0.001383); Jaewon Ind 2014 (0.457927); HwaseungR&A 2013 (0.007660) |
| Hanhwa | Decreasing | | Hanhwa 2011 (1.182940) |
| Hanhwa Advanced materials | Increasing | | Hanhwa 2011 (0.059321); HwaseungR&A 2013 (0.391092) |
| Hanhwa Chemical | Decreasing | | Namhae Chemical 2013 (4.996734); Hanhwa 2011 (0.115686) |
| Hyundai EP | Increasing | | Samnam petrochemical 2011 (0.014120); HyundaiEP 2013 (0.813576); HwaseungR&A 2013 (0.086734) |
| Hwaseung R&A | Increasing | | Hanhwa 2011 (0.043678); HwaseungR&A 2013 (0.489630) |
| Hwaseung Ind | Increasing | | Namhae Chemical 2011 (0.255379); Namhae Chemical 2013 (0.071045); HyundaiEP 2013 (0.665327) |
| Hyucamps | Increasing | | Namhae Chemical 2011 (0.541026); Polymirae 2011 (0.057508); Korea stirolution 2011 (0.185203) |

In this study, among the 55 firms, 11 firms were registered as the reference set. In particular, Hanhwa in 2011 and Hwaseung R&A in 2013 showed the best performance among these 11 reference sets, being reported 23 and 24 times as reference set, respectively. Their input and output mix were remarkably efficient from the perspective of green growth. This outstanding performance could be due to some financial or environmental achievements. For instance, Hanhwa was one of the leading firms in Korean petrochemical industry. Apparently, this leading firm already adopted well in ETS and accumulated know-how for sustainable management. Hanhwa also became the first Korean firm that implemented environmental management [34], an evidence showing its outstanding environmental performance. Furthermore, it has maintained its leading position in the Korean petrochemical industry, and this advantage seems to be reflected on its environmental performance too. Most benchmarking leaders did not keep their leading positions over time, implying the fragility and unsustainability of this position. In order to enhance sustainable governance, the government's more proactive regulations and more transparent and predictable pathways toward the 2030 ambitious target should be emphasized.

## 5. Conclusions and Discussion

In 2015, the Korean government inaugurated its ETS; indeed, evaluating the feasibility of the government's ETS policies may be too early because it is just four years from implementation. However, Korea is one of the world's largest emitting countries, and by the 2030s, government must reduce the emission by 37% from BAU. Therefore, it is an urgent task to evaluate E&E performance from the perspective of firms participating in ETS. In this study, we focused on petrochemical industry, which has the largest share as a sole industry. Our findings and policy implications are summarized as follows.

First, GTE value obtained from this study shows that Korean petrochemical industry needs to enhance its own efficiency. The average GTE score of Korea was 63.5%, implying a 36.5% potential for enhancement of GTE in Korea. Thus, firm leaders need to try their best to restructure their input mix for this enhancement. In particular, right after the implementation of ETS, the lowest GTE values were obtained, which indicates instability and failure of the regulation regime in motivating firms to participate. However, this negative trend changed positively (uptrend) from 2017, showing "J-curve effect". Therefore, we might have captured the evidence to anticipate sustainable future of Korean petrochemical industry under ETS. However, to maintain this uptrend, ETS should be sustainable from three aspects: profitability, growth capacity, and stability of ETS. In 2015, ETS was not workable because of lack of profitability coming from low carbon price. It was perceived as additional cost burden for firms. Therefore, governmental policy makers need to find more appropriate ways to increase carbon price and motivate firms to participate in ETS proactively. Once profitability problem is cleared, more firms are expected to participate in ETS, which will have positive effect on the entire sustainability.

Second, on the basis of the 2017 data, this study classified 39 and 16 firms under the IRS and DRS conditions, respectively. The 39 firms can enhance efficiency by enlarging their input mix, and the 16 firms can enhance their GTE by reducing their over-supplied input scale. This result implies that more investments do not always lead to an increase in GTE. To suggest more detailed guidelines for input variable mix, this study shows benchmarking information for each firm. Eleven firms are registered in the reference set in 2017. Since these firms are efficient in input mix, the rest of the firms that show similar scale should provide best performance in order to achieve calculated input target value. Eventually, this benchmarking process will be helpful for each firm in enhancing GTE. However, in order for these leading benchmarking firms to maintain their leadership, the Korean government should exhibit and maintain the transparent, yet predictable, pathways toward the very ambitious goal of reducing GHG emission until 2030.

Although this study has contributions from many perspectives, it still has several limitations. On the one hand, this study only focuses on petrochemical industry, which is insufficient in providing in-depth explanation about the impact of ETS because it consists of diversified sectors. Therefore, incorporating and analyzing the data from other sectors are recommended. Moreover, this study

cannot support statistical reliability; therefore, the bootstrapping method and SFA are required to overcome this shortage.

**Author Contributions:** The authors contributed to each part of the paper by: conceptualization, Y.C.; methodology, H.S.L.; software, H.S.L.; validation, Y.C.; formal analysis, H.S.L.; investigation, Y.C.; resources and data curation, Mastur; writing—original draft preparation, H.S.L.; writing—review and editing, Y.C.; visualization, H.S.L.; supervision, Y.C.; project administration, Y.C.; funding acquisition, Y.C.

**Funding:** This study is supported by the National Research Foundation of Korea (grant number NRF-2019R1A2C1005326).

**Conflicts of Interest:** The authors declare no conflict of interest.

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
