# Peer review of "Are Sustainable Development Policies Really Feasible? Focused on the Petrochemical Industry in Korea"

_sustainability, doi:10.3390/su11143980_

Round 1
Reviewer 1 Report
The paper’s title match its content. The research topic presented clearly, aim of article clearly specified and realized. The article have a logical layout. The language of article correct. The paper’s conclusions follow logically from the development of the argument. The text adequately illustrated.
A few comments to the authors below:
- line 107 - there is a "(GHG or CO2)" and there should be "(GHG, including CO2)" - CO2 is after all one of GHGs
- in the text in abbreviation "CO2" should be used subscript for "2" as in table 1;
- the source under table 1 should be on the same page as the table;
- line 221 - "The GTE scores range from 0.5 to 1, and the average GTE score is approximately 63.5%" - where do these percentages come from? It should be explained.
- the title of table 3 should be on the same page as the table
- the paper is not clear whether the abbreviation GTE means "greenhouse gas efiiciency" (as in the abstract) or "Green technical efficiency" (as in the Empirical results section and Fig. 2). It should be explained.
- the References section should be unified (including item No. 2 in the list).
Author Response
Thanks for your important comment, your comment was helpful obviously to enhance paper’s quality.
The paper’s title match its content. The research topic presented clearly, aim of article clearly specified and realized. The article have a logical layout. The language of article correct. The paper’s conclusions follow logically from the development of the argument. The text adequately illustrated.
A few comments to the authors below:
- line 107 - there is a "(GHG or CO2)" and there should be "(GHG, including CO2)" - CO2 is after all one of GHGs
Response: I changed it as you mentioned.
- in the text in abbreviation "CO2" should be used subscript for "2" as in table 1;
Response: We changed CO2 to CO₂.
- the source under table 1 should be on the same page as the table;
Response: We moved source to the same page, right below table 1.
- line 221 - "The GTE scores range from 0.5 to 1, and the average GTE score is approximately 63.5%" - where do these percentages come from? It should be explained.
Response: We changed paragraph as follow. Since the best efficiency is 1, 63.5% comes from the average score 0.635.
è During the sample period, the GTE scores of 55 firms range from 0.5 to 1, and the average GTE score of them is 0.635, approximately 63.5%.
- the title of table 3 should be on the same page as the table
Response: We moved title on the same page.
- the paper is not clear whether the abbreviation GTE means "greenhouse gas efiiciency" (as in the abstract) or "Green technical efficiency" (as in the Empirical results section and Fig. 2). It should be explained.
Response: GTE means greenhouse gas technical efficiency obviously and we revised expression in empirical result section.
- the References section should be unified (including item No. 2 in the list).
Response: We unified style of reference section.
Reviewer 2 Report
When you say: "South Korea (hereafter Korea) is one of the largest emitter countries" you need to justify in the bibliography.
The source of Figure 1 is missing.
Figure 2 is impossible to read. In addition, the caption should have a consistent styile, according to the journal guidelines.
When you say:"Although numerous methodologies and models are available for environment-friendly sustainability", you neeed to add the source/s. The same for the sentence: "For environmental research, two types of distance functions are widely used in the literature".
At page 13, Figure 2 should be Fugire 3 and the source is missing.
Athough the sample is limited and focuses only petrochemical industry, I strongly believe the methodology and, therefore, the results are interesting for the international community.
Author Response
Thanks for your important comment, your comment was helpful obviously to enhance paper’s quality.
The source of Figure 1 is missing.
Response: Since Figure 1 is drawn by author, we mentioned it.
è Figure 1. Korean BAU target by 2030 (Drawn by author)
Figure 2 is impossible to read. In addition, the caption should have a consistent style, according to the journal guidelines.
Response: 1. We revised Figure 2 more readable.
2. We rewrote caption based on journal publication style.
When you say: "Although numerous methodologies and models are available for environment-friendly sustainability", you need to add the source/s. The same for the sentence: "For environmental research, two types of distance functions are widely used in the literature".
Response: We added sources for following paragraphs.
è Although numerous methodologies and models are available for environment-friendly sustainability, the distance function approach has obtained much popularity because it provides a more flexible analysis of the joint-production technology with good product and environmental bad outputs simultaneously [2].
è For environmental research, two types of distance functions are widely used in the literature. They are Shephard distance function [4] and directional distance function (DDF) [5] respectively.
At page 13, Figure 2 should be Figure 3 and the source is missing.
Response: We corrected this as follows.
àFigure 3. GTE score from 2011 to 2017 (In average)
And Figure 3 is just shows annual GTE trend and it is drawn by author, so add paragraph as follow.
è The results are presented in Table 3, and Figure 3 shows annual GTE score’s average trend.
Although the sample is limited and focuses only petrochemical industry, I strongly believe the methodology and, therefore, the results are interesting for the international community.